# OpenReview forum: "COMMA: A Communicative Multimodal Multi-Agent Benchmark"
_TMLR — Accepted by TMLR_

### Review · Reviewer_Pdtk · 2025-06-12

**Summary Of Contributions:**

this paper looks at multiple LLM/VLM agents which interact (communicate) so as to solve a collection of some "toy" tasks such as operating an ATM or solving puzzles, specifically, one "solver" agent hich might be seen as the classic LLM/VLM agent, and one "expert" agent which i imagine to be a proxy for tools such as database access. the paper provides abstract description of the benchmark, link to code, and experimental evaluation of current LLM/VLM models on those tasks, and analysis of the errors/limitations of these models on the benchmark, and the reasons for these limitations. things such as distribution of errors types, how conversation length relates to success ratio, and token sequence length to performance is studied.

**Audience:**

Yes

**Broader Impact Concerns:**

-

**Claims And Evidence:**

Yes

**Requested Changes:**

consider improving the above points (making clearer which real-world tasks the benchmark represents; giving a more concrete picture/example of the (multi-agent) process that is studied; making more clear the GPT judge's role right at the beginning).

minor: i suggest to go again over which citation type (citet vs. citep) should be used where. e.g., "... even simple tests can effectively measure cognitive ability Davidson et al. (2006) ..." should probably be "... even simple tests can effectively measure cognitive ability (Davidson et al., 2006) ...".

**Strengths And Weaknesses:**

overall, this is a valuable contribution for benchmark-based understanding of LLM/VLM/"agentic AI" problems. also, i found it very interesting to see the comparison with the human on such things as efficiency score (amount of communication needed versus performance on the task goal). i think quite a diversity of toy tasks is contained in the benchmark. overall the paper is well-understandable.

weaknesses/ideas for improvement:

what was not so clear to me is how the benchmark relates to the specific real-world tasks that are currently important, such as LLM agents using databases, web browsers etc., because in the benchmark, all components are L/VLM agents, while in those real-world tasks, one component is usually a tool like database etc. it would be helpful if the authors could discuss a bit more of which concrete real-world LLM/VLM/"agentic AI" tasks the benchmark is representative, and of which not. the authors do briefly mention tasks like software development (i guess this refers to the "copilots" for coding), and this does in deed seem to be quite related to the proposed benchmark, but it would be worth going a bit more into the details of how features of those real-world tasks match features of the benchmark.

what i'm also missing a bit is a clearer picture of how specifically the whole process of interaction works, with all those components of agents, memory, communication. i think it would be very helpful to depict one example run of the whole interaction (either in main text or in appendix). i guess it's an autoregressive process where the agents generate the next tokens, but it remains unclear, how the call of memory is done, or how the handover between agents is done etc. is this all via one sequence of tokens, or are there other input/output possiblities for the agents, or? i also wouldn't mind an abstract, concise model of the whole interaction (is this something like multi-agent Markov decision process?). but sometimes the abstract model gets so involved that a simple example depiction can be easier. i see that figure 1 already somewhat depicts the whole process abstractly, but given this "stationary" depiction, i find it still hard to understand the concrete process. also i find the examples in figure 2 already helpful, but there, e.g., it remains unclear how the images are used (are they simply embedded and put into the same sequence as the text token embeddings?). but maybe i'm just missing the part of the paper where this is described.

my understanding is that the evaluation of errors etc. eventually was made by GPT? the authors do compare the GPT evaluation to human ones, so there is a validation of this approach. but it only seems done on a small dataset (10 representative examples for each case). therefore, there is uncertainty of how well generally the GPT judge works, while this working is central for the papers conclusion of model performances. maybe this reliance on the GPT judge and the limitation this implies can be made a bit more explicit right from the beginning. if i see correctly, currently this GPT-based aspect of the evaluation is not contained in abstarct or intro, but i might be mistaken.

---

> ### Author Response · Authors · 2025-07-04
> **Response to Review**
>
> Thank you for your thoughtful review and feedback! We are glad you found our contribution to agentic AI understanding valuable, and we have summarized our key draft modifications and responses to your request changes below:
>
> **How do the puzzles map onto real-world tasks:**
>
> We have expanded our discussion on the relevance of our tasks to real-world systems in our design principles section (3.1). In particular, we highlight similarity with coding copilot agents and multimodal tutoring systems, where one agent may have access to a visual interface (e.g., IDE) and another relies only on language instructions. Additionally, our setting closely resembles agentic workflows involving external tool use. For instance, one customer support dialog agent is giving instructions to guide another agent in using tools, e.g., querying databases, invoking APIs, or navigating a website.
> While the puzzle may seem simplistic, we believe they are intentionally designed to isolate and evaluate a fundamental capability: communication between AI agents. In any multi-agent system, effective communication is essential to coordinate actions, aggregate information, and achieve shared goals. To isolate this capability, we intentionally use simpler games that can be customized to probe this fundamental skill in our benchmark.
>
> **Better explanation of the whole process of interaction:**
>
> We appreciate the request for greater clarity of the process, and we have added an example full dialog in the appendix. We also refer the reader to this example in the caption for Figure 1.
>
> To clarify, each agent is an autoregressive model that predicts token sequences. Each agent is also a multimodal model capable of processing images as part of its input. If images are part of the input (e.g., solver puzzle image), the image is tokenized and concatenated with the instructions and conversation history. Importantly, our framework is model-agnostic, supporting any vision-language model that can take in an input image, conversation history (text), and the instruction prompt (text), and output a predicted action (text).
>
> **Limitations of GPT-Judge:**
>
> Our main results are scored automatically via rule-based evaluation. Each puzzle has a set of rules defined in the manual, which is provided to the expert. Only when the solver agent makes a mistake that does not pass the rule-based evaluation, we then use GPT judge to classify the type of mistake into one of four predefined categories for the purpose of error analysis. In other words, the GPT evaluator is not used to identify whether the solver agent made a mistake or not.
>
> We fully agree that LLM judges are not 100% reliable. However, since our core accuracy metrics rely on deterministic, rule-based evaluation, we believe the GPT categorization (used for Figure 3’s error breakdown) does not affect the overall validity of our results. Nevertheless, we now state the limitations of GPT-based judging more explicitly in Section 5.2 and have also included a note in the Broader Impact section to reflect this.

---

> > ### Comment · Reviewer_Pdtk · 2025-07-04
> >
> > thanks for your additions to the paper and explanation of the process.
> >
> > i'm not sure if this is already somewhere in the paper, but if not, i'd recommend to write a paragraph like the one in your response ("To clarify, each agent is an autoregressive model that predicts token sequences ... , and the instruction prompt (text), and output a predicted action (text).") in the paper, possibly the appendix if the main part is already too crowded.
> >
> > also i'd strongly recommend one more thing: currently, neither the abstract nor the bullet point list of contributions in the introduction contains a concrete description what the benchmark actually consists of, namely those puzzles. i feel, in the spirit of TMLR, it is important that claims match evidence, and in particular that the claims (in abstract/introduction) are specific enough. so i recommend to at least add 1-2 sentences in the abstract and introduction, that give a bit a concreter picture of what the benchmark consists of, namely those puzzles that aim at covering certain cognitive abilities. this should give an idea of the area that is captured by those puzzles, as well as its limitations.

---

> > > ### Author Response · Authors · 2025-07-05
> > > **Follow up to Reviewer Comment**
> > >
> > > Thank you for your suggested changes! We have added a paragraph explaining the details of the agent interaction to the appendix, in the section which illustrates the full dialog. The "Agent Architecture" paragraph of the "Design Principles of the Benchmark" section in the main text also refers to this appendix when we describe the agent interaction.
> > >
> > > We have also modified the abstract and 1 bullet point in the introduction contributions to emphasize the point that our benchmark evaluates LLM performance using puzzles. Here are the revised 2 sentences in the abstract:
> > >
> > > To fill this gap, we introduce COMMA: a novel **puzzle benchmark** designed to evaluate the collaborative performance of multimodal multi-agent systems through language communication. **Our benchmark features a variety
> > > of multimodal puzzles**, providing a comprehensive evaluation across four key categories of
> > > agentic capability in a communicative collaboration setting.

---

> > > > ### Author Response · Authors · 2025-08-11
> > > > **Friendly Follow-up**
> > > >
> > > > Dear Reviewer Pdtk,
> > > >
> > > > We sincerely thank you for your thoughtful and detailed review of our manuscript. Your positive feedback is greatly appreciated, and we are particularly encouraged by your remarks that our work makes a valuable contribution to benchmark-based understanding of LLM, VLM, and agentic AI problems. We are excited that you found the comparison with humans both interesting and informative, especially regarding the efficiency score metric. We are also delighted that you believe our paper is clear and understandable, and that our benchmark contains a large diversity of tasks.
> > > >
> > > > **We are writing to follow up and confirm whether our responses have fully addressed your concerns, or if there are any additional questions we can clarify. We want to ensure that every point you raised has been carefully considered.**
> > > >
> > > > Once again, thank you for your time, effort, and valuable contributions to strengthening our work!
> > > >
> > > > Best regards,
> > > >
> > > > Authors

---

### Review · Reviewer_HXsk · 2025-06-12

**Summary Of Contributions:**

This paper introduces COMMA (Communicative Multimodal Multi-Agent Benchmark), a new benchmark designed to assess how well multimodal AI agents can collaborate and communicate using language, particularly when they have incomplete or unequal information access. The key contributions and new knowledge include:
(1)  COMMA simulates real-world collaborative tasks, inspired by games like "Keep Talking and Nobody Explodes," where agents (a "Solver" and an "Expert") must communicate effectively to solve puzzles with differing information access.
(2) The evaluation reveals surprising weaknesses in leading multimodal models, including proprietary ones like GPT-4o and reasoning models like o4-mini, as well as open-source models.
(3) The study categorizes common failure modes in agent communication, such as misinterpretation, role play errors, and repeat loops, providing clear examples and insights for future research.
(4) COMMA specifically evaluates agents' ability to handle private information, showing that even high-performing models like GPT-4o and Gemini 2.0 often disclose sensitive data despite instructions to withhold it, highlighting a critical area for improvement in AI agent interaction.
(5) The benchmark introduces detailed metrics including Success Rate (SR), Partial Success Rate (PSR) for multi-step tasks, Efficiency Score (balancing success and conciseness), Average Mistakes (AM), and Average Conversation Length (ACL).

**Audience:**

Yes

**Broader Impact Concerns:**

The ethical implications of the work on the COMMA benchmark, which would necessitate a Broader Impact Statement, largely center on the Privacy and Sensitive Data Handling - The benchmark includes puzzles like "ATMPuzzle" and "TelehealthPuzzle" that specifically involve private financial or health information. The findings indicate that even high-performing models (like GPT-4o and Gemini 2.0) frequently disclose private details despite explicit instructions to withhold them, while models like o4-mini adhere more closely to privacy constraints but at the cost of lower task performance. This highlights a critical challenge for deploying multimodal agents in real-world scenarios, particularly in healthcare where privacy concerns are paramount.

A Broader Impact Statement would need to discuss:
- The inherent risks of using and further developing models that struggle with privacy adherence when handling sensitive user data.
- The potential for data leakage or unauthorized access if such agents are deployed in applications involving confidential information.
- The responsibility of researchers and developers to prioritize robust privacy-preserving mechanisms before real-world deployment, emphasizing that benchmark success rates do not guarantee privacy in practice

**Claims And Evidence:**

No

**Requested Changes:**

Proposed Adjustment to the submission:

(1) The current human evaluation is described as a "preliminary study" with the authors role-playing agents due to cost and time constraints. This significantly limits the generalizability and direct comparability of the human baseline. I would recommend to secure acceptance, the authors should either:
-  Design and execute a more extensive human study with external participants to validate the human baseline and provide a more robust comparison point for the AI models. This would significantly strengthen the ecological validity of the benchmark.
- Stronger Justification for Current Limitations - If a larger human study is truly infeasible, the authors need to provide a much more compelling and detailed argument for why the current limited human evaluation is sufficient for the claims made, and clearly outline concrete plans for future work to address this limitation.

**Strengths And Weaknesses:**

Strengths:

(1) The paper effectively highlights and addresses the significant oversight in current multimodal agent research regarding collaborative communication, unequal information access, and tasks beyond individual capabilities. This is a crucial area for real-world AI deployment.
(2) COMMA is a well-conceived benchmark inspired by cooperative gameplay (like "Keep Talking and Nobody Explodes") and cognitive science principles. It uniquely focuses on language-based communication and collaborative problem-solving between agents with distinct information access.
(3) The findings expose unexpected struggles in state-of-the-art models, including powerful proprietary ones (like GPT-4o and Gemini 2.0 Flash) and reasoning models (like o4-mini). Many AI models, particularly open-source reasoning models, perform poorly in collaborative communication, sometimes even worse than a random baseline.
(4) The benchmark specifically evaluates the ability of agents to withhold sensitive information, revealing that even high-performing models frequently disclose private details despite explicit instructions. This identifies a critical area for future research.

Weaknesses:
(1) The authors state that human participation was a "preliminary study" and that they "role-played as agents ourselves" due to cost and time constraints. While understandable, this limits the generalizability of the human baseline and the direct comparison with AI models. More extensive human-AI interaction studies would strengthen the ecological validity.
(2)  The paper acknowledges an "inevitable simulation-to-reality gap". While it's important to state this, a more detailed discussion of specific challenges in bridging this gap for this benchmark and potential strategies for future work to address it would be beneficial.
Scope of Puzzle Scenarios: The paper mentions that experiments "may not represent all possible scenarios in our puzzles"  and that a "more comprehensive evaluation of puzzle categories" is left for future work. This suggests that the current set of puzzles, while varied, might not fully capture the breadth of collaborative multimodal tasks, which could affect the comprehensiveness of the evaluation.
(3) "Random" Baseline Performance: It's concerning that some state-of-the-art models struggle to outperform even a simple random agent baseline in agent-agent collaboration. While this highlights a growth area, a deeper analysis of why this occurs and what specific architectural or training limitations contribute to this poor performance compared to a random baseline would be insightful.

---

> ### Author Response · Authors · 2025-07-04
> **Response to Review**
>
> Thank you for your thoughtful review and feedback! We are glad you think our work addresses a crucial area for AI deployment, and we have summarized our key draft modifications and responses to your request changes below:
>
> **Changes:**
>
> We have significantly changed the broader impact statement to discuss the privacy and data-sensitive handling and other limitations as requested. More concretely:
>
> **Need for more human baselines:**
>
> We agree that a larger, unbiased human sample size would strengthen the validity of our reported performance. However, our current codebase is primarily designed to support automatic evaluation in a large number of agent-agent interaction trials in a local server, thus it poses practical challenges for scaling up to external participants on crowdsourcing platforms (like Amazon Mechanical Turk), as it requires participants to set up a special conda environment, and most Mechanical Turk workers aren’t familiar with this. Additionally, each worker can’t repeat each puzzle too many times without the risk of introducing bias. We have added a discussion in the broader impact and limitations section to discuss this aspect of our work.
>
> Despite these constraints, we believe that our current human evaluation still provides a useful signal, which is an upper bound of human performance under idealized conditions. While average performance from a broader pool of external participants may be lower, our results nevertheless reveal consistent trends in human responses. First, our current human baseline reflects a tendency for human communication to be more concise than that of language models, a pattern likely to persist even after the inclusion of more human solver trials. Second, our human participants show nearly perfect vision perception and describe image content with high precision.
>
> To improve this human baseline, several aspects can be considered in future work. First, the codebase can be redesigned for web-based deployment and crowdsourcing a diverse pool of participants. Second, in addition to random puzzle initialization, the implementation of randomized rules in the instruction manual may help curate unbiased human performance metrics. We will add this discussion to the draft.
>
> **Bridging the simulation reality gap:**
>
> We have added a section to discuss this in our draft in the broader impact + limitations section (reproduced here for convenience):
>
> Similar to other benchmarks, our framework necessarily simplifies real-world complexity. While our experiments involve sampling puzzle configurations and simulating multi-turn agent conversations, they do not exhaust the vast space of all possible collaborative interactions. For instance, real-world scenarios often involve collaboration with more than 2 people and hierarchical role structures in corporate environments. To help bridge this gap, future work may consider adding more realistic collaboration environments. First, implement a hierarchical structure of agent communication, such as a workplace of multiple agents for developing software. Second, incorporating real-world modality data (e.g., images, audio, or video) may help simulate a more faithful multimodal setting. For example, interacting with medical imaging software would be much more complicated than our current puzzles, but simulate a realistic scenario. Lastly, the inclusion of more human-in-the-loop evaluation of performance can provide more accurate insight.
>
> **Random baseline analysis:**
>
> We added clarification on the random baseline behavior and discussion about its competitive performance compared with LLaVA-CoT and R1-OneVision. We summarize the discussion here for convenience:
>
> - **Random Baseline:** This baseline selects a random action uniformly from a pool of valid actions at each timestep. It does not utilize any information from the expert agent, puzzle state, conversation history, or past actions.
>
> Interestingly, the random baseline performs on par with models like LLaVA-CoT and R1-OneVision, despite having no access to the puzzle state or expert knowledge. We found that long CoT models such as LLaVA-CoT attempt to solve the puzzles independently through chain-of-thought reasoning, but they struggle to generalize to the communication setting introduced in our benchmark. These models are typically optimized for single-agent, instruction-following tasks, and their reasoning often fails to align with the implicit coordination required between agents. In contrast, the random baseline naturally explores a broader set of trajectories by sampling uniformly from the space of valid actions. This stochasticity allows it to occasionally stumble upon a correct solution. As a result, the random baseline can match or even outperform models that rely on self-reasoning strategies in unfamiliar collaborative contexts.

---

> > ### Author Response · Authors · 2025-08-11
> > **Friendly Follow-up**
> >
> > Dear Reviewer HXsk,
> >
> > We would like to express our sincere gratitude for your thorough and insightful review of our manuscript. We greatly appreciate your positive feedback, and we were especially encouraged by your remarks noting that our benchmark addresses a critical area in multimodal agentic AI. Your validation of our work has been both motivating and invaluable.
> >
> > **We would like to kindly follow up to confirm whether our responses have fully addressed your comments, or if there are any remaining questions we can clarify. We are eager to ensure that all of your points have been carefully considered.**
> >
> > Thank you once again for your time and your valuable contributions to this process!
> >
> > Best regards,
> >
> > Authors

---

> > > ### Comment · Reviewer_HXsk · 2025-08-11
> > > **Respomse**
> > >
> > > Yes - your responses fully address my comments. You are good to move forward.

---

### Review · Reviewer_H6co · 2025-06-28

**Summary Of Contributions:**

The paper introduces a novel benchmark specifically designed to evaluate the collaborative capabilities of multimodal agents through language-based communication. The authors assess how multiple agents interact, share asymmetric information, and work together to solve vision-language puzzles. They model a structured interaction between two agents, a “Solver” with visual access to a task and an “Expert” with only instruction knowledge, who must communicate via natural language to complete a shared objective.
The benchmark includes 10 task types spanning four categories of cognitive skills: memory recall, multimodal grounding, multi-step reasoning, and handling of private information. The authors evaluate a broad set of state-of-the-art proprietary and open-source models (e.g., GPT-4o, Gemini, LLaVA, o4-mini), revealing that most models struggle with effective inter-agent communication.
To better understand these limitations, the paper introduces a taxonomy of common failure modes role confusion, repetition loops, miscommunication, and misinterpretation, and conducts both qualitative and quantitative analyses using a GPT-based judge.

**Audience:**

Yes

**Broader Impact Concerns:**

1. The benchmark includes puzzles involving sensitive data (e.g., health information or bank PINs). While this is simulated, it implicitly encourages testing agents in privacy-critical scenarios, and current models, particularly GPT-4o and Gemini, are shown to frequently violate privacy constraints. This highlights an ethical gap in current model behavior.
2. The paper uses GPT-4o as an automated judge to evaluate GPT-4o and similar models. This creates a conflict of interest or model class circularity, where evaluation may be biased in favor of the models being tested.
3. As noted earlier, the human baseline was established by the authors themselves, who also designed the tasks. This raises concerns of unconscious bias, which may misrepresent the true gap between humans and AI agents.
The authors should expand the Broader Impact section to include:
- A stronger caution regarding the risks of applying collaborative agents in high-stakes or regulated domains.
- Ethical considerations around using language-only communication to simulate sensitive decision-making.
- Transparency about evaluation bias risks, especially when using model-based judges
- Declaration of the use of a small, potentially biased human sample, composed of the authors, should be made more explicit in the manuscript

**Claims And Evidence:**

Yes

**Requested Changes:**

1. Please further clarify the comparison to random baseline: the claim that models such as LLaVA-CoT and R1-OneVision "struggle to outperform even a simple random agent baseline" needs careful contextualization.
- Is the poor performance is due to: failure in communication ability, repetition loops, or noise/randomness in output?
- Please add the behavior of the random baseline more explicitly. I see the definition in the caption but you may want to add it in the main text.
2. The paper uses a GPT-based judge for failure mode classification but only briefly mentions calibration. There are risks of bias, subjectivity, or circularity, especially since GPT-4o is also being evaluated.
- Please provide more detail on the calibration process.
- Include a discussion of possible biases (e.g., same-model judging).
3. The paper reports human performance on 100 sampled puzzles as a reference baseline. However, it states that the authors themselves played the roles of solver or expert due to the cost of external participants. This introduces a risk of strong experimenter bias, as the authors may already know puzzle structures, solutions, or design intent, consciously or unconsciously influencing their performance.
- Clearly acknowledge the potential for bias in the human results and treat the reported human baseline as an upper bound rather than an objective gold standard.
- Suggest recruiting external annotators or conducting blinded evaluations in future iterations to establish a more robust human benchmark.

**Strengths And Weaknesses:**

Strengths:
1. The paper is inspired by real-world human-agent cooperation, with asymmetric information settings. The analogy to "Keep Talking and Nobody Explodes" is clever and effective.
2. Incorporation of private information handling, multimodal grounding, memory recall, and multi-step reasoning provides a holistic challenge for models.
3. Failure mode taxonomy (role play errors, repetition loops, miscommunication, misinterpretation) is thoughtfully curated.
4. Open-sourcing of data and code is mentioned, supporting future research replication and extension.

Weaknesses:
1. The paper is using a GPT-based judge to perform analysis on failure mode. If the GPT-based judge is architecturally or behaviorally similar to the evaluated agents (e.g., GPT-4o as solver or expert), it may be biased toward excusing its own errors or interpreting ambiguous outputs more favorably. Additionally, judging behavior depends heavily on the prompting strategy used. If the judge prompt is vague, underspecified, or overly leading, it may misclassify or hallucinate failure types. Even the authors note they had to “make minor adjustments” to calibrate the GPT-judge, this subjectivity introduces reproducibility risks.
2. The paper could benefit from an ablation study to show how specific features (e.g., episodic memory input or asymmetric views) affect performance.
3. Human Baseline Evaluation Process (See point 3 below for additional clarification).

---

> ### Author Response · Authors · 2025-07-04
> **Response to Review**
>
> Thank you for your thoughtful review and feedback! We are glad you think our work addresses a crucial area for AI deployment, and we have summarized our key draft modifications and responses to your request changes below:
>
> **Changes:** \
> We have significantly changed the broader impact statement to discuss the privacy and data-sensitive handling and other limitations as requested. More concretely:
>
> **Need for more human baselines** \
> We agree that a larger, unbiased human sample size would strengthen the validity of our reported performance. However, our current codebase is primarily designed to support automatic evaluation in a large number of agent-agent interaction trials in a local server, thus it poses practical challenges for scaling up to external participants on crowdsourcing platforms (like Amazon Mechanical Turk), as it requires participants to set up a special conda environment, and most Mechanical Turk workers aren’t familiar with this. Additionally, each worker can’t repeat each puzzle too many times without the risk of introducing bias. We have added a discussion in the broader impact and limitations section to discuss this aspect of our work.
>
> Despite these constraints, we believe that our current human evaluation still provides a useful signal, which is an upper bound of human performance under idealized conditions. While average performance from a broader pool of external participants may be lower, our results nevertheless reveal consistent trends in human responses. First, our current human baseline reflects a tendency for human communication to be more concise than that of language models, a pattern likely to persist even after the inclusion of more human solver trials. Second, our human participants show nearly perfect vision perception and describe image content with high precision.
>
> To improve this human baseline, several aspects can be considered in future work. First, the codebase can be redesigned for web-based deployment and crowdsourcing a diverse pool of participants. Second, in addition to random puzzle initialization, the implementation of randomized rules in the instruction manual may help curate unbiased human performance metrics. We will add this discussion to the draft.
>
> **Bridging the simulation reality gap**
>
> We have added a section to discuss this in our draft in the broader impact + limitations section (reproduced here for convenience):
>
> Similar to other benchmarks, our framework necessarily simplifies real-world complexity. While our experiments involve sampling puzzle configurations and simulating multi-turn agent conversations, they do not exhaust the vast space of all possible collaborative interactions. For instance, real-world scenarios often involve collaboration with more than 2 people and hierarchical role structures in corporate environments. To help bridge this gap, future work may consider adding more realistic collaboration environments. First, implement a hierarchical structure of agent communication, such as a workplace of multiple agents for developing software. Second, incorporating real-world modality data (e.g., images, audio, or video) may help simulate a more faithful multimodal setting. For example, interacting with medical imaging software would be much more complicated than our current puzzles, but simulate a realistic scenario. Lastly, the inclusion of more human-in-the-loop evaluation of performance can provide more accurate insight.
>
> **3. Random baseline analysis:**
>
> We added clarification on the random baseline behavior and discussion about its competitive performance compared with LLaVA-CoT and R1-OneVision. We summarize the discussion here for convenience:
>
> - **Random Baseline:** This baseline selects a random action uniformly from a pool of valid actions at each timestep. It does not utilize any information from the expert agent, puzzle state, conversation history, or past actions.
>
> Interestingly, the random baseline performs on par with models like LLaVA-CoT and R1-OneVision, despite having no access to the puzzle state or expert knowledge. We found that long CoT models such as LLaVA-CoT attempt to solve the puzzles independently through chain-of-thought reasoning, but they struggle to generalize to the communication setting introduced in our benchmark. These models are typically optimized for single-agent, instruction-following tasks, and their reasoning often fails to align with the implicit coordination required between agents. In contrast, the random baseline naturally explores a broader set of trajectories by sampling uniformly from the space of valid actions. This stochasticity allows it to occasionally stumble upon a correct solution. As a result, the random baseline can match or even outperform models that rely on self-reasoning strategies in unfamiliar collaborative contexts.

---

> ### Author Response · Authors · 2025-07-04
> **(Real) Response to Review**
>
> We apologize for the previous response meant for a different reviewer! Openreview is not allowing us to modify/delete comments.
>
> Thank you for your thoughtful review and feedback! We are glad you found the analogy to the Keep Talking game clever, and we have summarized our key draft modifications and responses to your request changes below:
>
> **Suggested ablations:**
>
> We currently have an ablation in Figure 4 in which the maximum conversation length, and hence episodic memory, is limited. However, we believe the suggested ablations are interesting ideas worth exploring and are actively working to run some additional experiments which we hope to finish very soon:
> - An experiment where the solver and expert do not have access to past conversation history in their prompts
> - An experiment in which a single agent attempts to solve the puzzle with full information
> Regarding the suggestion for an ablation with asymmetric views, we wanted to clarify if this refers to a setting in which both agents have full access to the information? If so, we believe the single agent ablation will have a similar result.
>
> **Random baseline clarification:**
> We added clarification on the random baseline behavior and discussion about its competitive performance compared with LLaVA-CoT and R1-OneVision. We repeat the discussion here for convenience:
>
> - **Random Baseline:** This baseline performs a uniformly random valid action at each timestep. It does not utilize any information from the expert agent, puzzle state, or conversation history.
>
> Interestingly, the random baseline performs comparably to models like LLaVA-CoT and R1-OneVision, despite having no access to the puzzle state or expert knowledge. We found that long CoT models such as LLaVA-CoT attempt to solve the puzzles independently through chain-of-thought reasoning, but they struggle to generalize to the novel communication setting introduced in our benchmark. These models are typically optimized for single-agent, instruction-following tasks, and their reasoning often fails to align with the implicit coordination required between agents. In contrast, the random baseline naturally explores a broader set of trajectories by sampling uniformly from the space of valid actions. This stochasticity allows it to occasionally stumble upon a correct solution. As a result, the random baseline can match or even outperform models that rely on self-reasoning strategies in unfamiliar collaborative contexts.
>
> **More detailed calibration process + highlight biases present:**
>
> We have added details in section 5.2 about the GPT-Judge calibration process and its limitation. Specifically, we first identified common failure cases we were observing (e.g., miscommunication) that prevented the solver from finishing the puzzle or making progress. We then manually and randomly sampled 10 instances of each failure in the conversations as well as 10 random normal conversation logs. The calibration process involved continuously refining the prompt used for the GPT-Judge until it achieved a high accuracy. We then used this prompt along with the GPT judge to evaluate all conversations.
>
> We fully acknowledge that this is not a perfect system, and there may be bias present in the form of the GPT-judge favouring its own response. We have attempted to make the definitions in the prompt as clear as possible to mitigate this bias. Importantly, we also wish to highlight that the GPT-Judge is not our primary metric for measuring accuracy performance. The main table results are derived from rule-based scoring and task progression metrics. The goal of the GPT-Judge is to complement the error analysis of our benchmark in a scalable manner. We have also added a section in our broader impact statement to acknowledge this.
>
> **Acknowledging bias in human baseline:**
>
> We kindly refer the reviewer to the previous response, or the response to reviewer HXsk, for our discussion regarding the human baseline results.

---

> > ### Author Response · Authors · 2025-08-11
> > **Friendly Follow-up**
> >
> > Dear Reviewer H6co,
> >
> > We sincerely thank you for your thoughtful and detailed review of our manuscript. Your positive feedback is greatly appreciated, and we are particularly encouraged by your observations on the cleverness of our analogy to the Keep Talking game and our simulated collaborative settings. Your recognition of these aspects has been both motivating and deeply rewarding for our team.
> >
> > **We are writing to follow up and confirm whether our responses have fully addressed your concerns, or if there are any additional questions we can clarify. We want to ensure that every point you raised has been carefully considered.**
> >
> > Once again, thank you for your time, effort, and valuable contributions to strengthening our work!
> >
> > Best regards,
> >
> > Authors

---

### Author Response · Authors · 2025-07-14
**Follow-Up on Rebuttal and Updated Draft**

We would like to thank the reviewers for their thoughtful feedback and suggestions! We are especially excited that reviewer H6co found our analogy to the Keep Talking game clever, and reviewers HXsk and Pdtk believe that our work addresses a critical area in multimodal agentic AI. If there are any remaining concerns or suggestions in response to our updated draft or rebuttal, we would be grateful for your feedback. Thank you all again for your time and consideration!

---

### Decision · Action_Editor_hJ4B · 2025-08-11

**Recommendation:** Accept with minor revision

**Additional Comments:**

The authors should submit a camera ready version with all promised changes.

**Audience:**

Yes

**Audience Explanation:**

I think anyone interested in probing cooperation and engagement between modern models will find this benchmark useful and interesting.

**Claims And Evidence:**

Yes

**Claims Explanation:**

This paper introduces COMMA, a benchmark designed to evaluate how multimodal, multi-agent systems collaborate using language. The COMMA benchmark features a variety of multimodal puzzles to test agents across four categories of collaborative capability.

The authors provide results showing significant weaknesses in state-of-the-art models, including SoTA models like GPT-4o and reasoning models like o4-mini. As well, they show that chain-of-thought models such as R1-Onevision and LLaVA-CoT struggle to perform better than a random agent baseline in agent-to-agent collaboration.

The reviewers were in agreement that the results provide convincing enough evidence that the COMMA benchmark is an useful test to assess shortcomings in today's large models in multi-agent, communicative settings. Given the consensus amongst the reviewers, an accept decision is natural.